# Study on the Preparation and Properties of Bridge Concrete Using Low Carbon Aggregates

**DOI:** 10.3390/ma16010245

**Published:** 2022-12-27

**Authors:** Ruishuang Jiang, Youjia Xing, Shuai Liu, Yongzhi Guo, Baolin Guo

**Affiliations:** 1Shandong Transportation Institute, Jinan 250031, China; 2Jinan Kingyue Highroad Engineering Co., Ltd., Jinan 250101, China; 3Key Laboratory of In-Service Bridge Performance Evaluation and Promotion, Jinan 250031, China; 4Shandong Engineering Research Center of Concrete Materials and Bridge Structures, Jinan 250031, China

**Keywords:** low carbon aggregate, bridge concrete, mechanical properties, anti-permeability, frost resistance

## Abstract

It is an outstanding solution for protecting the environment using manufactured sand instead of natural river sand in concrete. In this paper, tunnel granite muck was processed into low carbon, coarse and fine aggregates, and low carbon aggregates were used to prepare bridge concrete. Meanwhile, the mechanical properties, anti-permeability, and frost resistance of concrete were investigated. The results demonstrated that the concrete prepared using low carbon aggregate had higher mechanical properties than concrete prepared using river sand. The chloride ion penetration resistance of concrete using low carbon aggregate is better than that of concrete using river sand, and frost resistance has been improved.

## 1. Introduction

With the rapid decline of natural sand resources in the main rivers and lakes, the enhancement of ecological protection, and the restriction of mining through prohibitive mining policies, some highway and railway infrastructure have processed tunnel muck according to local conditions into mechanical sand aggregate, which is used to formulate concrete for tunnels and bridges [1,2,3]. It effectively solves the contradiction between the shortage of natural sand resources and the difficulty of disposing of excess tunnel muck.

The utilization of tunnel muck could reduce the disposal of solid waste, reduce the effect of muck on farmland and river courses, alleviate the destruction of mountains and vegetation caused by stone material collection, and maintain the integrity of the ecological landscape. At the same time, it is an effective measure when using electric dust collection or washing methods to reduce environmental pollution. Bellopede et al. [4] thought tunnel muck would be widely used and bring significant environmental and economic benefits through proper management and treatment. Gertsch et al. [5] believe that because tunnel muck is not fully understood, it is rarely used in engineering construction. In fact, if they conform to building aggregate standards, the hard rock dregs can be used for highway concrete and structural concrete.

As one of the important components of concrete, the quality of fine aggregate has an important influence on the working properties of a concrete mixture and the mechanical properties of hardened concrete. Natural river sand as a high-quality concrete fine aggregate has been nearly exhausted, and its exploitation destroys the ecological balance. In this situation, concrete producers began to use manufactured sand to gradually replace natural river sand to solve the problem of building sand. In recent years, many scholars have learned about the physical properties of manufactured sand [6,7,8], the mechanism of sand grading [9,10,11], its mechanical properties, and the durability of concrete prepared using granite manufactured sand. Martins et al. [12] used manufactured sand to replace all-natural river sand and analyzed the influence of sand’s compressive strength using mechanics. The results prove that, compared with natural sand, the compressive strength of manufactured sand is higher than that of concrete composed of natural sand at the same water–cement ratio. Westerholm et al. [13,14] studied the influence of manufactured sand on the fluidity of mortar, and mechanical sand has an important influence on the rheological results and compressive strength of mortars. Xie Kaizhong et al. [15] found that the curves of the concrete slump, the cubic compressive strength, the axial compressive strength, the splitting tensile strength, and the elastic modulus of the different manufactured aggregate samples increased and then decreased.

At present, more and more road and bridge engineering projects are using manufactured sand to make concrete structures or concrete cement pavements. This study uses manufactured sand instead of river sand to prepare concrete and explores the difference between the concrete’s mechanical properties and durability properties through testing the concrete’s mobility energy, compressive strength, flexural strength, tensile strength, elastic modulus, and chlorine ion permeability. This study effectively solves the shortage of aggregate and has an important guiding significance for the utilization of tunnel muck.

## 2. Materials and Methods

### 2.1. Materials

Cement: the cement was ordinary Portland cement P·O 42.5 from Shandong Shanshui Cement Group Co., Ltd. (Jinan, Shandong, China), and its basic properties are shown in Table 1.

Fly ash: grade II fly ash was purchased from Chiping Xinyuan environmental protection building materials Co., Ltd. (Liaocheng, Shandong, China), with a burning loss of 3.4% and water requirement ratio of 103%.

Fine aggregate: Granite manufactured sand and river sand were used as the coarse aggregate in the experiment. The basic properties of the fine aggregate are shown in Table 2.

Coarse aggregate: Granite and limestone manufactured stone were used as the coarse aggregate in the experiment. The crushed stone was composed of 5–10 and 10–20 grades and the mixing ratio was 3:7. Among them, the granite gravel water absorption was 0.8%, mud content was 1.0%, and the crushing value was 12.5%. Crushed limestone water absorption was 0.6%, mud content was 0.6%, and the crushing value was 10.5%.

### 2.2. Mix Proportion

The concrete mix proportions are illustrated in Table 3.

### 2.3. Test Method

#### 2.3.1. Workability

The workability of the concrete was measured according to GB/T 50080-2016 (standard for test methods of performance on ordinary fresh concrete).

#### 2.3.2. Mechanical Properties

The compressive strength (specimen size: 100 mm × 100 mm × 100 mm, 3 specimens for each age period of 5 d, 7 d, 28 d, and 90 d providing a total of 12 specimens), flexural strength (specimen size: 100 mm × 100 mm × 400 mm, 3 specimens for each age providing a total of 12 specimens), splitting tensile strength (specimen size: 150 mm × 150 mm × 150 mm, 3 specimens for each age providing a total of 12 specimens), and elastic modulus (specimen size: 150 mm × 150 mm × 300 mm, 6 specimens for each age providing a total of 24 specimens) of concrete were measured according to GB/T 50081-2019 (standard for test methods of concrete’s physical and mechanical properties). All specimens were cured for 24 h. After curing for 24 h, the molds of the specimens were removed, and the specimens were kept in a concrete curing room (temperature 20 °C ± 2 °C and relative humidity greater than 95%). After curing the specimens for 5 d, 7 d, 28 d, and 90 d, compressive strength, flexural strength, splitting tensile strength, and elastic modulus were tested, respectively.

#### 2.3.3. Chloride Ion Permeability Resistance

The electric flux and RCM method were recorded in accordance with GB/T 50082-2009 (standard for test methods of long-term performance and durability of ordinary concrete). The cylinder specimens (100 mm diameter × 50 mm length) were prepared, with 3 specimens for each age. In total, 6 specimens were needed for the electric flux and 6 specimens were needed for the RCM method. After being cured for 28 d and 180 d, the chloride ion permeability resistance of the concrete specimens was tested using the electric flux and RCM method.

The ability of the concrete samples to resist chloride ion penetration was evaluated using the RCM method of JGJ/T 193-2009 (standard for inspection and assessment of concrete durability) and the NEL method of CCES01-2004 (guide to durability design and construction of concrete structures).

#### 2.3.4. Freeze–Thaw Resistance

The freeze–thaw resistance of concrete was investigated according to GB/T 50082-2009 (standard for test methods of long-term performance and durability of ordinary concrete). After being cured for 28 days, three concrete specimens (100 × 100 × 400 mm) were frozen for 4 h and thawed in water for 4 h. After 25 freeze–thaw cycles, the relative dynamic modulus of elasticity of the concrete specimens was tested. After 300 freeze–thaw cycles, the experiment was stopped. Concrete is frost-resistant when its relative dynamic modulus of elasticity decrease is lower than 60% after n cycles.

#### 2.3.5. The Air-Void Parameters

The air bubble characteristic parameters in hardened concrete were tested according to JTJ 270-98 (water transport engineering concrete test code). Tests were performed using three polished concrete specimens 100 × 100 × 20 mm cut from cube specimens. Results of measurements were available as a set of standard parameters for air void microstructure characterization:− Spacing factor L (mm).− Specific surface α (mm^−1^).− Air content A (%).

## 3. Results and Discussion

### 3.1. Workability

The working performance of the concrete prepared for the tests is shown in Figure 1. The flowability tests on the three groups of concrete show that slight bleeding occurs in the GM mixture group, the flowability and viscosity of the GR mixture group are satisfactory, and the LM mixture group is sticky and has poor mobility. The result shows a high positive correlation between the slump constant and air content in the concrete mixture. When the amount of water reducer is the same, the slump of the concrete prepared using natural sand is up to 210 mm, and the air content is 2.2%, while the slump of the concrete prepared using the manufactured sand is 185 mm and 170 mm, and the air content is 1.6% and 1.2%.

It is very clear that the use of manufactured sand reduces the slump of the concrete mixture, which can be attributed to the physical properties of manufactured sand. Visually, the particle shape of manufactured sand is more angular than natural sand. There is a higher friction force among the particles, weakening the lubrication effect of mortar and thus lowering the flowability of the fresh concrete. The lithology of virgin rock, the surface texture of particles, and defects caused by mechanical crushing of manufactured sand determine the adsorption capacity of the particle surface on the adsorption water reducer and affect the fluidity of concrete [16]. Furthermore, the lithology and the particle shape of manufactured sand influence the water absorption performance of manufactured sand. Basically, the rheological behavior of concrete depends on water film thickness on the surface of manufactured sand and cementing material particles [17].

In addition, the use of manufactured sand reduces the air content of concrete because it is difficult for air bubbles to remain in the manufactured sand mortar. The workability of the concrete was further reduced by decreasing the air content. Thus, manufactured sand affects the workability of fresh concrete through its characteristics and the changes of the air content in fresh concrete.

### 3.2. Mechanical Properties

Figure 2 displays the results of the compressive strength and bending strength of concrete. With increasing age, the compressive strength and flexural strength of concrete prepared with manufactured sand and natural sand, respectively, increase gradually, and the rate of the strength increase appears rapidly at first, and slows after the curing age of 28 days. The compressive strength of the GM group is the highest, the GR group is second best, and the GM group is the lowest. Comparing the experimental results between the GM and GR groups shows that the compressive strength of concrete prepared using manufactured sand is higher than concrete prepared using natural sand. The compressive strength of GM was 7.8% higher than that of GR when the curing age is 5 d. At 90 d, the compressive strength of GM was 14.6% higher than that of GR.

The strength of concrete prepared using low carbon aggregate is thought to be largely correlated with the bond of manufactured sand and cementing materials. Compared with natural sand, manufactured sand is more firmly embedded in pastes, mainly because the surface of manufactured sand has a crushed, fractured surface, which is beneficial to the contact between manufactured sand and pastes. The stone powder with a particle size less than 75 um that comes from the mechanical crushing of virgin rock contributes to filling the voids in concrete. Furthermore, the main component of rock is CaCO_3_, which can react with Ca(OH)_2_ generated by cement hydration, resulting in an increase in the cohesive force of the mixture [18,19].

Comparing the experimental results between the GM and GR groups shows that the compressive strength of concrete prepared with granite aggregate is higher than that with limestone aggregate, and the compressive strength of GM is 9%~14% higher than LM at each age. Compared with limestone, the granite aggregate used has higher water absorption, which reduces the water–binder ratio in the interface transition zone, improves the cohesiveness between the slurry and the aggregate, and improves the compressive strength of the granite concrete. In addition, under the same stress level, granite has greater deformation, which can alleviate the effect of stress concentration, thereby improving the compressive strength of concrete.

The flexural strength of concrete increases with age; however, the flexural strength of concrete prepared with manufactured sand is basically equivalent to that with natural sand. Comparing the experimental results of the GM and GR group, the manufactured sand has no significant effect on the flexural strength. The experimental results of the GM and LM group show that the flexural strength of the concrete made of granite aggregate is higher than that of the limestone aggregate. As the age increases, the difference of strength between the two tends to decrease.

The splitting tensile strength of concrete prepared using low carbon aggregate is shown in Figure 3. The specimens in the GR group show higher split tensile strength. On the contrary, the specimens in the GM group have the lowest split tensile strength. Manufactured sand reduced the tensile strength of the concrete. Compared to the specimens in the GR group, the tensile strength of the specimens in the GM group decreased by 15.8% and 20.4% from 5 d to 7 d. With the increase of the curing age, the tensile strength difference between the GM and GR group specimens gradually decreased.

The elastic modulus of concrete prepared with manufactured sand and with natural sand is shown in Figure 4. Comparing the elastic modulus of the GR and GM groups shows that manufactured sand will reduce the elastic modulus of concrete, but the reduction is limited. With the increase of the curing age, the elastic modulus of the specimens in the two groups is roughly the same. It is worth noting that the elastic modulus of concrete prepared with granite aggregate is significantly lower than that with limestone aggregate. The elastic modulus of specimens in the GM group is 26.3% lower than those in the LM group concrete at 5 d. With the increase of the curing age, this gap did not decrease. Because the granite aggregate used in this experiment has a low elastic modulus, and coarse aggregates have a vital influence on the elastic modulus of concrete, the use of aggregates with low elastic modulus directly leads to concrete with a lower elastic modulus.

### 3.3. Chloride Ion Permeability Resistance

As shown in Figure 5, the ability to resist chloride ion penetration in concrete prepared with manufactured sand and with natural sand at 28 d and 180 d was characterized by the RCM method and the electric flux method. The ability to resist chloride ion penetration in specimens in the GM group, whose specimens were prepared with manufactured sand, is better than those in the GR group prepared with natural sand, especially at the age of 28 days when the improvement effect is more obvious. This is because the manufactured sand contains a certain amount of stone powder, which can fill the pores in the matrix, refine the pore structure of the concrete matrix, increase the density of the matrix, and reduce the intrusion of chloride ions. Comparing the results of the GR and GM group shows that the ability to resist chloride ion penetration in the concrete made of granite and limestone were basically the same. The electric flux is less than 1200 C, and the chloride ion diffusion coefficient is less than 1.0 × 10^−12^ m^2^/s.

### 3.4. Frost Resistance and Air-Void Parameters

The frost resistance of concrete with different aggregates is shown in Figure 6. With increasing the freeze–thaw cycle times, the relative dynamic modulus of elasticity of all the concrete samples decreases gradually. The relative dynamic modulus of elasticity of the LM sample drops to 58% after 150 times, indicating that the LM sample has poor frost resistance. However, the GM and GR specimens show a higher relative dynamic modulus of elasticity after 300 times. The results indicate that granitic manufactured sand could produce concrete with excellent frost resistance, which means that the frost resistance of concrete has little connection to the type of fine aggregate.

Air-void structure is the key parameter that affects the frost resistance of concrete using research results of the publication [20,21]. The air void characteristics of concrete with different aggregates are shown in Table 4. The results show that a higher air content in concrete has better frost resistance. In addition, with increasing air content, the spacing coefficient of hardened concrete decreases, which is helpful to improve the frost resistance of concrete. It can be interpreted using Powers’ hydrostatic model [22]; the expansion stress decreases as the space between air voids decreases since it shortens the distance of capillary water to the free surface of air voids.

## 4. Conclusions

(1)Low carbon aggregates from tunnel muck affects the workability of fresh concrete through its characteristics and the changes in air content in fresh concrete. Therefore, the mix ratio should be optimized to obtain concrete prepared using low carbon aggregate with a better working performance in actual engineering applications.(2)The GM specimens show higher compressive strength. At 90 d, the compressive strength of GM was 14.6% higher than that of GR. The flexural strength of concrete prepared with manufactured sand is basically equivalent to that with natural sand. Compared to the GR specimens, the splitting tensile strength of GM decreased by 15%~20%. In addition, low carbon aggregates from tunnel muck reduces concrete’s elastic modulus because of the low elastic modulus of coarse granite aggregate; the elastic modulus of the GM specimens is 26.3% lower than those of LM at 5 d.(3)The ability to resist chloride ion penetration in concrete prepared using manufactured sand is better. Compared with river sand (GR), the electric flux and chloride ion diffusion coefficient of the specimens with granite manufactured sand (GM) decreased by 30% and 24.6%, respectively.(4)Concrete specimens with low carbon aggregates show better frost resistance properties. The concrete with the limestone aggregate (LM) could withstand the freeze–thaw cycle 150 times; however, the concrete samples with low carbon aggregates (GM or GR) could withstand the freeze–thaw cycle 300 times, indicating that low carbon aggregates can be used in special service environments.(5)In actual projects, low carbon aggregates prepared from tunnel muck can be used to prepare high-strength, high-performance, and durable concrete that meets the requirements of the project by controlling the quality of the granite manufactured sand and selecting a reasonable mix ratio.

## Figures and Tables

**Figure 1 materials-16-00245-f001:**
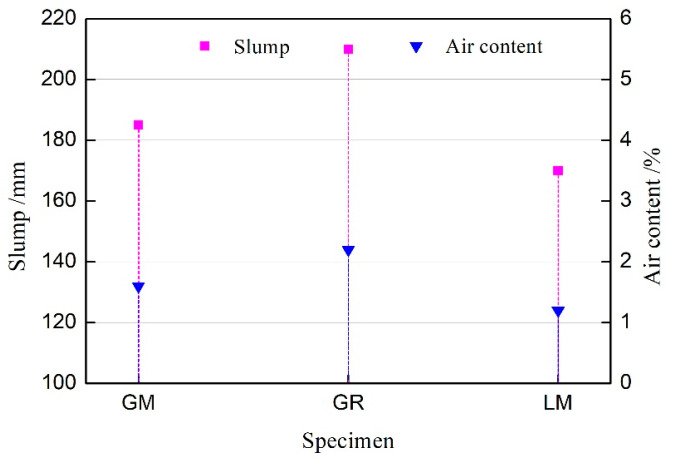
Slump and air content of the concrete mixture.

**Figure 2 materials-16-00245-f002:**
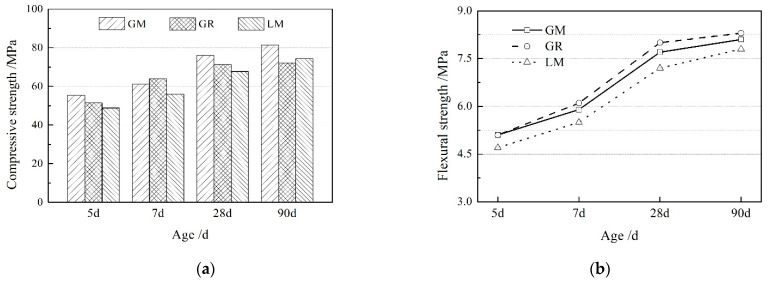
The mechanical strength of concrete prepared using low carbon aggregate: (**a**) compressive strength and (**b**) flexural strength.

**Figure 3 materials-16-00245-f003:**
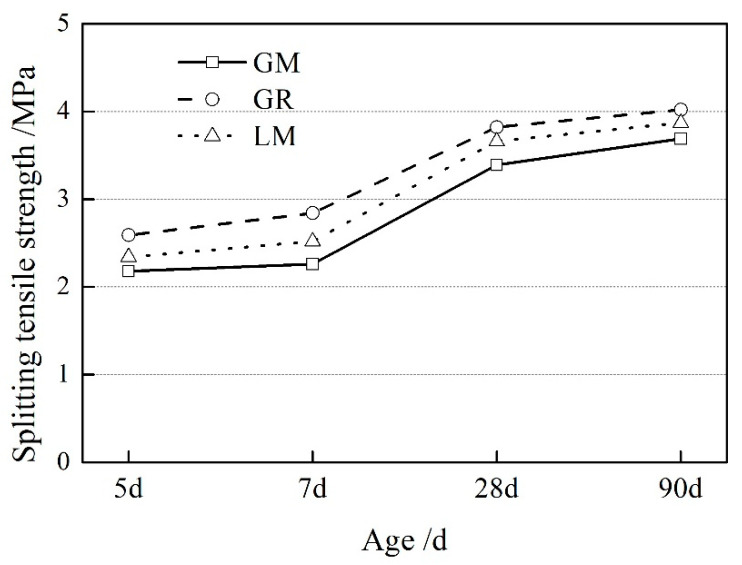
The splitting tensile strength of concrete prepared using low carbon aggregate.

**Figure 4 materials-16-00245-f004:**
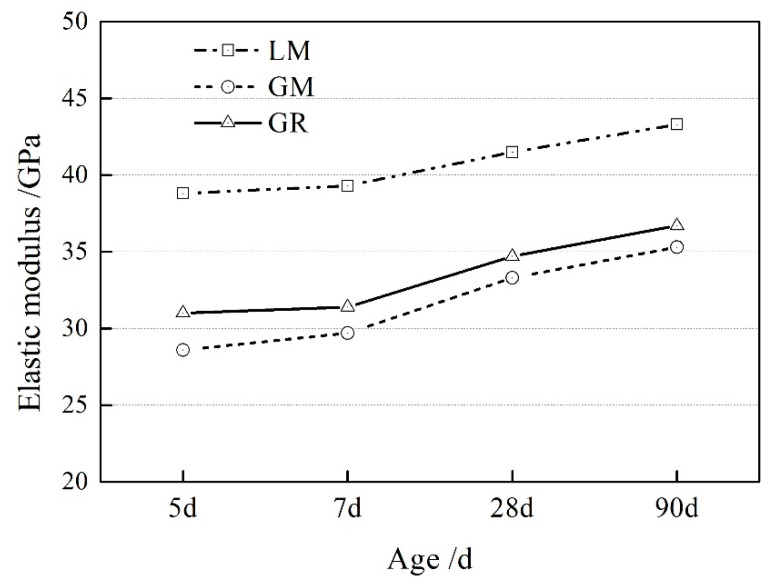
The static elastic modulus of concrete prepared using low carbon aggregate.

**Figure 5 materials-16-00245-f005:**
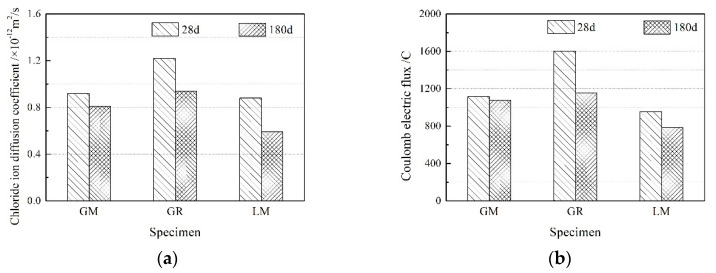
Chloride ion permeability resistance of concrete prepared using low carbon aggregate: (**a**) the RCM method and (**b**) the electric flux method.

**Figure 6 materials-16-00245-f006:**
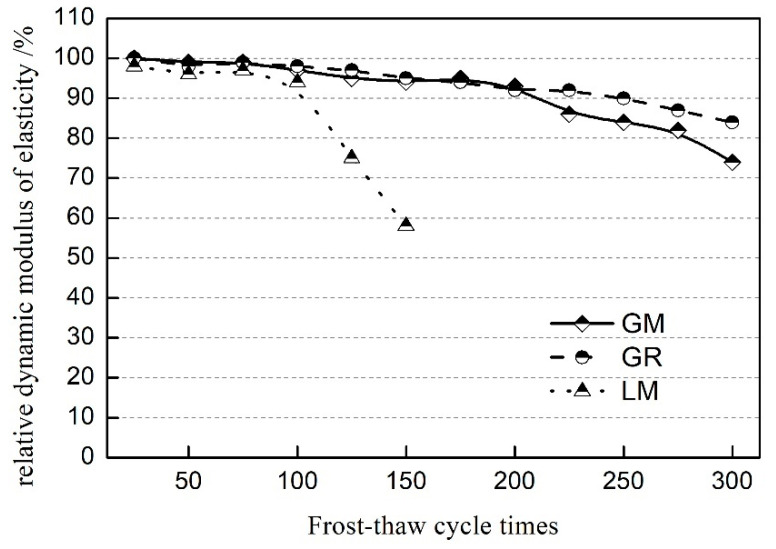
Relative dynamic modulus of elasticity of specimens.

**Table 1 materials-16-00245-t001:** Basic properties of P·O 42.5 cement.

Standard Consistency Water Consumption/%	Setting Time/h		Flexural Strength/MPa		Compressive Strength/MPa		SO_3_ Content/%	Other Substances/%
	Initial	Final	3 d	28 d	3 d	28 d		
29.5	4.75	5.83	4.6	6.8	20.0	43.6	1.96	1.88

**Table 2 materials-16-00245-t002:** Physical properties of granite manufactured sand and river sand.

Type	Accumulated Screen Residue/%						Fineness Modulus	<75 μm Content/%	Water Absorption/%
	4.75	2.36	1.18	0.6	0.3	0.15			
Granite sand	0.9	6.3	28.0	37.9	85.1	97.8	2.53	3.7	1.46
River sand	8.5	23.9	37.3	62.7	88.4	94.4	2.89	1.1	1.20

**Table 3 materials-16-00245-t003:** Mix ratio of concrete with granite manufactured sand and river sand.

Sample	Coarse Aggregate	Sand	Content (kg/m^3^)					
			Cement	Fly Ash	Sand	Aggregate	Water	Water Reducer
GM	Granite	Granite sand	420	50	700	1050	158	5.2
GR	Granite	River sand	420	50	700	1050	158	5.2
LM	Limestone	Granite sand	420	50	700	1050	158	5.2

**Table 4 materials-16-00245-t004:** The air-void characteristics of specimens.

Specimens	A (%)	α (mm^−1^)	L (mm)
GM	1.7	17.29	0.920
GR	2.4	28.79	0.553
LM	1.0	3.23	1.140

## Data Availability

The data presented in this study are available upon request from the corresponding author.

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
