# Peer review of "Study on the Preparation and Properties of Bridge Concrete Using Low Carbon Aggregates"

_materials, 2022, doi:10.3390/ma16010245_

Round 1

Reviewer 1 Report

The Authors compared the physical and chemical properties of concrete prepared from tunnel granite cave slag and processed into low-carbon coarse and fine aggregate.

Acceptable methods were used to test the chemical physical properties: chloride ion permeability resistance, freeze-thaw resistance and the air-voids parameters.

Question: why the compressive strength test specimens were not 150mm x 150mm x 150mm.

Positive remarks

1. The Authors present the research results very clearly.

2. The Authors do not show more references than is needed in this paper.

Negative remarks

1. The tests were performed for one class of concrete. The results are thus qualitative, not quantitative.

2. I couldn't find information about the number of specimens.

3. The Authors write that this is an excellent environmental solution by using artificial sand instead of natural river sand in the concrete.

But the Authors do not write anything about how the processing of tunnel granite cave slag into low-carbon coarse and fine aggregate affects the environment.

Reviewer 2 Report

In this paper, the respected authors have investigated properties of concrete materials made of low carbon aggregate. They have used manufactured sand instead of river sand to prepare concrete. This paper is written well. However, the novelty and contribution of the present paper is not clear. Many papers have previously been published which deal with properties of concrete made of low carbon aggregate.

Additional comments (materials-2078928)

1) In Fig. 1, the workability of GM, GR and LM samples are compared with each other. However, the air contents of these samples are not the same. It is well-known that by increasing the air content, the workability of the concrete increases. Please discuss about this issue more in the manuscript.

2) The compressive strength of concrete at 28th day has usually marginal discrepancy with the compressive strength of the concrete at 90th day. However, Fig. 2a shows that the compressive strength of the GM samples at 28th day is larger than that the compressive strength at 90th day. Why?

3) Splitting tensile strength and flexural strength of the concrete samples are both indirectly related to the tensile strength of the concrete material. However, results of Fig. 2b and Fig. 3 shows different trends for GM, GR and LM samples. In Fig. 2b, LM samples have lowest flexural strength while in Fig. 3, the GM samples have the lowest splitting tensile strength. What is the reason?

4) The conclusion part should be supported by data. For example, please add in this section that how much the compressive strength of concrete samples increases if manufactured sand is used instead of river sand? 

Round 2

Reviewer 1 Report

The current version is acceptable.

Merry Christmas & Happy New Year.

Reviewer 2 Report

The paper is recommended for publication at the present format.